# Joint Feature-Space and Sample-Space Based Heterogeneous Feature Transfer Method for Object Recognition Using Remote Sensing Images with Different Spatial Resolutions

**DOI:** 10.3390/s21227568

**Published:** 2021-11-14

**Authors:** Wei Hu, Xiyuan Kong, Liang Xie, Huijiong Yan, Wei Qin, Xiangyi Meng, Ye Yan, Erwei Yin

**Affiliations:** 1Tianjin Artificial Intelligence Innovation Center (TAIIC), Tianjin 300457, China; huvivi325@163.com; 2China Great Wall Industry Corporation, Beijing 100000, China; kongxiyuan88@126.com; 3Defense Innovation Institute, Academy of Military Sciences (AMS), Beijing 100000, China; xielnudt@gmail.com (L.X.); xdqinwei@126.com (W.Q.); mengdeersijiu@163.com (X.M.); yinerwei1985@gmail.com (E.Y.); 4National Key Laboratory of Human Factors Engineering, China Astronaut Research and Training Center, Beijing 100000, China; 18610881830@126.com

**Keywords:** heterogeneous feature transfer, classification of remote sensing images, transfer learning, negative transfer

## Abstract

To improve the classification results of high-resolution remote sensing images (RSIs), it is necessary to use feature transfer methods to mine the relevant information between high-resolution RSIs and low-resolution RSIs to train the classifiers together. Most of the existing feature transfer methods can only handle homogeneous data (i.e., data with the same dimension) and are susceptible to the quality of the RSIs, while RSIs with different resolutions present different feature dimensions and samples obtained from illumination conditions. To obtain effective classification results, unlike existing methods that focus only on the projection transformation in feature space, a joint feature-space and sample-space heterogeneous feature transfer (JFSSS-HFT) method is proposed to simultaneously process heterogeneous multi-resolution images in feature space using projection matrices of different dimensions and reduce the impact of outliers by adaptive weight factors in the sample space simultaneously to reduce the occurrence of negative transfer. Moreover, the maximum interclass variance term is embedded to improve the discriminant ability of the transferred features. To solve the optimization problem of JFSSS-HFT, the alternating-direction method of multipliers (ADMM) is introduced to alternatively optimize the parameters of JFSSS-HFT. Using different types of ship patches and airplane patches with different resolutions, the experimental results show that the proposed JFSSS-HFT obtains better classification results than the typical feature transferred methods.

## 1. Introduction

With the development of satellite sensor technology, more and more earth observation data with higher spatial resolution can be acquired from satellites for remote sensing image (RSI) analysis and processing, among which the classification of RSIs has long been a hot research topic due to its wide range of applications for both military and civil fields [1,2,3].

To build effective classification results for high-resolution images, it is essential to collect sufficient labeled samples of high-resolution images to ensure adequate training of the classifier, while it will incur a high cost for labeling a large number of samples. Instead of training the classifier using only labeled high-resolution images, one can consider combining previously labeled low-resolution images with currently labeled high-resolution images to train the classifier jointly. For the RSIs with different resolutions, the extracted features present different characteristics, and thus they can be represented mathematically as samples following the different distributions. Obviously, this violates the basic hypotheses of supervised learning, i.e., both the training data and test data are drawn from the same distribution [4]. Therefore, the classifier trained by RISs of different resolutions will fail to yield good classification results.

To solve this problem, one can take into consideration the introduction of transfer learning technology [5,6], which refers to transfer knowledge or experience available for one or more domains (i.e., source domain) to improve the performance for a new yet related domain (i.e., target domain). The source domain or target domain consists of three concepts, feature space X, marginal probability distribution P(X), and conditional probability distribution P(Y|X), where $X=\{x_{1},x_{2},\ldots ,x_{N}\}\in X$ and *Y* denotes the class labels. For traditional machine learning, it requires the training and test data to be represented in the same feature space and obey the same distribution P(X). In contrast, transfer learning allows the training and test data to be represented in different feature spaces and follow the various distributions. The transfer learning methods can be categorized into instance-oriented methods [7], feature-oriented methods [8], and classifier-oriented methods [9]. The instance-oriented methods assign different weights to the instances from source data to reduce the distribution difference between the source domain and target domain, e.g., [10]. The feature-oriented methods always utilize the feature projection mapping to search a subspace that can minimize the distribution distance, e.g., maximum mean distance (MMD) [11] and Hilbert–Schmidt independence criterion (HSIC) [12]. In addition, the classifier-oriented methods utilize samples from the target domain to fine-tune the model parameters learning from the source domain to improve the performance of the classifier for the target domain [9,10]. To solve the problem whereby RSIs of different resolutions present different feature distributions, this paper focuses on feature-oriented methods. Feature-oriented methods can be further categorized into two groups according to whether the data from the source domain and target domain are represented in the same feature space or not, i.e., homogeneous feature transfer and heterogeneous feature transfer. Most pioneered works focus on homogeneous feature transfer, including transfer component analysis (TCA) [12], joint distribution adaptation (JDA) [13], and structural correspondence learning [14]. Although these methods worked well for specific applications, they require data from the source domain and target domain to be characterized in the feature space with the same dimension and cannot deal with RSIs with the different resolutions because RSIs with different resolutions always present different dimensions of features. To deal with the case that samples from the source domain and target domain present different dimensions, heterogeneous feature transfer methods have been developed in the recent decade. For example, the heterogeneous domain adaptation method is built to utilize projection matrices with different sizes and a joint kernel regression model to learn shared features from different domains [15]. In addition, the domain adaptation manifold alignment method was developed to convert instances in each source domain into a common subspace through their respective mapping functions, and then the mapped instances in the common space are used to learn a linear regression model [16]. In order to reduce the difference of the conditional distribution between domains, heterogeneous feature augmentation (HFA) is constructed to use two transformation matrices to map both the source domain and the target domain into the common latent space and then minimize the structural risk function of the support vector machine [17].

For object recognition using RSIs, the object patches from different resolutions usually present different sizes, and they can be naturally described as features with different dimensions, denoted as a heterogeneous feature in this paper, and the detailed definition is as follows:

**Definition** **1.**
*(Heterogeneous features and heterogeneous feature transfer) Given two features describing objects, if these two features present different dimensions, they are considered heterogeneous features. In addition, if features from the source domain and features from the target domain present different dimensions, the corresponding transfer learning method is regarded as heterogeneous feature transfer.*


The traditional homogeneous feature transfer methods cannot deal with these data. Although the existing heterogeneous feature transfer methods can handle cases where samples from the source and target domain present different dimensions, they focus on feature space-based mapping to reduce the difference in distribution between the source domain and target domain, ignoring the difference between sample space between the source domain and target domain. In detail, the sample space in our paper is defined as a set $\Omega =\left\{S_{1},S_{2},\ldots ,S_{N}\right\}$, where the elements $S_{i}$ and *N* denote the *i*-th sample and the number of samples, respectively. If RSIs with poor quality exist, it is easy to cause the negative transfer. The negative transfer [18] means that the transferred results do not increase or even affect the learning performance of the target domain. Motived by the abovementioned factor, a joint feature space and sample space-based heterogeneous feature transfer method (JFSSS-HFT) is proposed in this paper to learn the projection matrices of different sizes to map heterogeneous samples from the source domain and target domain to the common feature space, meanwhile reducing the impact of the outliers from the sample space adaptively to avoid the occurrence of negative transfer.

The main contributions of the proposed method can be summarized as three factors.
Since image patches of objects collected from RSIs with different resolutions present different sizes, the extracted features from image patches with different resolutions should present different dimensions. In comparison to most of the existing feature transfer methods that utilize the same projecting matrix to deal with data from the source domain and target domain with the same dimension, the proposed JFSSS-HFT method constructs two projecting matrices with different sizes. This is so that it can map data with different dimensions to the common space to reduce the difference between domains, and it makes our JFSSS-HFT suitable for processing heterogeneous remote-sensing data.Compared with the existing methods that only focus on the feature-space-based mapping to reduce the difference between different domains, the proposed JFSSS-HFT jointly considers the feature space and sample space to select and map the features of representative samples to improve the effect of feature transfer and reduce the occurrence of negative transfer [18] caused by outlier samples.To achieve heterogeneous feature transfer by jointly considering feature space and sample space, the JFSSS-HFT method is proposed in this paper, and then the alternating-direction method of multipliers (ADMM) is introduced to solve the corresponding optimization problem. The experiment results demonstrate that the proposed JFSSS-HFT can obtain better classification results compared with typical feature transfer methods using RSIs with different resolutions and imaging angles.

The remainder of this paper is organized as follows. In Section 2, a review of the related work of the proposed method is given briefly. In Section 3, the JFSSS-HFT is presented, and the detailed solving method is described. In Section 4, the performance of the proposed JFSSS-HFT is evaluated compared with state-of-the-art feature transfer methods using airplane patches and ship patches with different resolutions. Our conclusion is given in Section 5.

## 2. Related Work

Before presenting our work, certain basic concepts related to transfer learning are briefly introduced.

Maximum mean discrepancy: Maximum mean discrepancy (MMD) [12], widely used in the transfer learning field, can measure the distance between two distributions (i.e., samples distributions in the source domain and target domain) in the reproducing kernel Hilbert space (RKHS) ℋ. Given samples from two domains $X^{S}={\left\{x_{i}^{S}\right\}}_{i=1}^{N_{S}}$ and $X^{T}={\left\{x_{i}^{T}\right\}}_{i=1}^{N_{T}}$, the empirical estimate of MMD between two domains is calculated by $MMD\left(X^{S},X^{T}\right)=\min_{\varphi :X\to H}\;{∥\frac{1}{N_{S}}\sum_{i=1}^{N_{S}}\varphi \left(x_{i}^{S}\right)-\frac{1}{N_{T}}\sum_{i=1}^{N_{T}}\varphi \left(x_{i}^{T}\right)∥}_{ℋ}^{2}$, where φ(⋅) and ${\left\|\cdot \right\|}_{ℋ}$ denote the function that maps the samples from original space X to RKHS ℋ and RKHS norm. Referring to [12], it is shown that MMD will asymptotically approach zero if and only if the two distributions are the same when the RKHS is universal. Note that the MMD is a nonparametric distance estimate. Compared with the Kullback–Leibler (KL) divergence relying on a priori knowledge of the probability density [19], MMD can be embedded into the feature transfer method conveniently. Therefore, many typical feature transfer methods adopt MMD as the distance measure, e.g., [12,13,18].

Feature transformation-based transfer learning: Classical supervised learning assumes that the samples from the training set and test set obey the same distribution. When the samples from the training set and test set present different distributions, the classical learning algorithms obtain poor results. To solve this problem, feature-transformation-based transfer learning technology was developed to map the samples from different domains into the common space to reduce the difference between domains. Most existing feature-transformation-based transfer learning methods achieve domain adaptation via maximizing the overlap between the transformed samples or minimizing the specific distances between the transformed samples (e.g., MMD) from different domains [13,18]. For example, the graph-based feature transformation methods project the original features to a subspace with a small dimension by maintaining the graph structure that describes the relationship between samples [20,21]. The joint distribution adaptation method minimizes the MMD as the objective function to adapt both marginal and conditional distributions between domains [13]. These feature transformation methods can reduce the difference between different domains while it learns the transformation function of all the samples. When samples with poor quality exist, e.g., RSIs obtained under poor imaging conditions, the performance of feature transfer will be decreased and cause negative transfer [18]. To reduce the impact of negative transfer, the active transfer learning method is constructed to remove samples that are too different from the normal samples during the transfer [18]. The drawback of this method is that it only considers the outliers in the source domain and ignores the outliers in the target domain, and it can only handle homogeneous cases, i.e., it requires the samples from the source domain and target domain to be located in the feature space with the same dimensions.

## 3. Joint Feature Space and Sample Space-Based Heterogeneous Feature Transfer Method

For image classification in remote sensing, although numerous remote-sensing images are increasingly available, the image patches containing different types of objects consistent with the resolution of the test data are expensive to collect, and the resulting insufficient training set is difficult to support effective training of the classifier. Instead of training the classifier using image patches with high spatial resolution, one can consider exploiting the image patches with different spatial resolutions to improve the classification performance for RSIs with high resolution. In this way, the high-resolution image patches are considered the target domain data, and the low-resolution image patches are considered the source domain data. The task aims to improve the classification results of the target domain by exploiting the data from the source domain. Note that traditional machine learning requires the training data and test data to obey the same distribution, while the source domain data and target domain data present a different data distribution. Therefore, feature transfer technology is needed to transfer the source domain data into the target domain to improve the classification performance in the target domain. In addition, image patches of objects with different resolutions always present different sizes, i.e., the extracted features for the source domain and target domain are located in feature space with different dimensions. Therefore, it is necessary to construct the heterogeneous feature transfer method to map features with different dimensions to the common space to reduce the difference between the source domain and target domain. Moreover, to prevent the image patches containing occlusion from causing negative transfer, compared to existing methods that only consider the projection mapping of the feature space, it is essential to avoid the occurrence of negative migration by analyzing the sample space to reduce the impact of outliers.

In this way, the JFSSS-HFT method is proposed to weigh standard samples adaptively and learn the heterogeneous projection function acting on samples from different dimensions to reduce the difference of distribution between the source domain and target domain. Applying the JFSSS-HFT method, the obtained heterogeneous projection function can be used to map heterogeneous data to the common sub-space and then train a classifier jointly to obtain an effective classification result. An illustration of the proposed JFSSS-HFT is shown in Figure 1.

As shown in Figure 1, the objective function of JFSSS-HFT contains three terms, i.e., maximizing the interclass variance term and the adaptive outlier eliminating term, and minimizing the MMD term. In addition, the main variables of JFSSS-HFT to be optimized include two projection matrices with different sizes that are used for mapping the input features from the source domain and the target domain and two adaptive weight factors of the source domain and the target domain that are used to evaluate the importance of each sample for transfer. Furthermore, to obtain the optimal variables of JFSSS-HFT, the ADMM method is constructed to solve the optimization problem of JFSSS-HFT. Based on the optimization solution, the mapped features from the different domains are considered the transferred results and can be used to train the classifier. Given a test image with a high resolution, the corresponding features are extracted and mapped into the common space, and then its class label can be predicted using the trained classifier.

### 3.1. Construct the Optimization Problem of JFSSS-HFT

For simplicity and without loss of generality, both the number of source domains DS and the number of target domains DT are set to one. Given source domain data, XS∈ℝMS×NS, where in each column in matrix XS, the *M*_S_ and *N*_S_ denote the sample in the source domain, the dimension of the feature, and the number of samples in the source domain, respectively. For target domain data, XT∈ℝMT×NT, where in each column in XT, *M_T_* and *N_T_* denote the sample in the target domain, the dimension of the feature, and the number of samples in the target domain, respectively. Note that MS may not be equal to MT, and NS may not be equal to NT. yiS∈{1,…,k} and yiT∈{1,…,k} represent the class label of the *i*-th sample in the source domain and the *i*-th sample in the target domain, respectively. To reduce the difference between transferred features from different domains, the MMD distance is adopted to evaluate the distribution difference between the source domain and target domain. To apply MMD as a proper measurement to reduce the distribution difference, two projection matrices with different dimensions are constructed to map features from the source domain and target domain to the common space, respectively, as shown in Equation (1).
(1)minPS,PT‖1NS∑i=1NSPSxiS−1NT∑i=1NTPTxiT‖F2
where xiS, xiT, PS∈ℝM×MS, and PT∈ℝM×MT denote the *i*-th sample from the source domain, the *i*-th sample from the target domain, the projection matrix (i.e., the projection matrix in Figure 1) associated with source domain data, and the projection matrix associated with target domain data, respectively. Note that the optimization problem only focuses on the marginal probability distribution between source domain data and target domain data, i.e., P(xS) and P(xT). For the labeled samples, compared with a marginal probability distribution, it is necessary to consider the conditional probability distribution to improve the classification performance after transfer [13], i.e., P(xSyS) and P(xTyT). Based on this idea, we construct the following optimization problem to minimize the MMD of the conditional probability distribution.
(2)minPS,PT∑y=1k‖1#{yiS=y|i}∑yiS=yPSxiS−1#{yiT=y|i}∑yiS=yPTxiT‖F2
where #{yiS=y|i} denotes the number of elements in the set. Since some samples may present poor quality due to partial occlusion or different imaging conditions, the traditional feature transfer methods that utilize all the samples easily cause negative transfer. To reduce the occurrence of negative transfer, similar to active transfer learning technology [18], the adaptive weight parameters are constructed and embedded into the optimization problem to filter the outlier in the sample space, as shown in Equation (3).
(3)minPS,PT,αiS,αiT∑y=1k‖1#{yiS=y|i}∑yiS=yPSxiSαiS−1#{yiT=y|i}∑yiS=yPTxiTαiT‖F2s.t.0≤αiS≤10≤αiT≤1∑iαiS=(1−wS)NS∑iαiT=(1−wT)NT
where αiS denotes the adaptive weight factor of the source domain (see Figure 1), αiT is the adaptive weight factor of the target domain (see Figure 1), wS is the pre-established proportion of outliers for the source domain, and wT is the pre-established proportion of outliers for the target domain. The bound constraints indicate the range of αiS and αiT is from 0 to 1. The larger αiS and αiT, the larger the weight of the corresponding sample used to affect the value of the projection matrix. If αiS or αiT is 0, the corresponding samples are considered outliers and are not used for feature transfer.

In addition, to improve the classification performance after transfer, based on the theory of linear discriminant analysis [22], the maximization of interclass variance is taken into consideration. In this way, the following optimization problem can be obtained.
(4)$$\begin{array}{c} \min_{P_{S},P_{T},\alpha_{i}^{S},\alpha_{i}^{T}}\;\sum_{y=1}^{k}{∥\frac{1}{\#\left\{y_{i}^{S}=y|i\right\}}\sum_{y_{i}^{S}=y}P_{S}x_{i}^{S}\alpha_{i}^{S}-\frac{1}{\#\left\{y_{i}^{T}=y|i\right\}}\sum_{y_{i}^{S}=y}P_{T}x_{i}^{T}\alpha_{i}^{T}∥}_{F}^{2}-\\ C\times \sum_{y=1}^{k}{∥\frac{1}{N_{y}}\left(\sum_{y_{i}^{S}=y}P_{S}x_{i}^{S}\alpha_{i}^{S}+\sum_{y_{i}^{T}=y}P_{T}x_{i}^{T}\alpha_{i}^{T}\right)-\frac{1}{N_{ST}}\left(\sum_{i}P_{S}x_{i}^{S}\alpha_{i}^{S}+\sum_{i}P_{T}x_{i}^{T}\alpha_{i}^{T}\right)∥}_{F}^{2}\\ s.t.\\ 0\leq \alpha_{i}^{S}\leq 1\\ 0\leq \alpha_{i}^{T}\leq 1\\ \sum_{i}\alpha_{i}^{S}=\left(1-w_{S}\right)N_{S}\\ \sum_{i}\alpha_{i}^{S}=\left(1-w_{T}\right)N_{T}\\ N_{ST}=\left(1-w_{S}\right)N_{S}+\left(1-w_{T}\right)N_{T}\\ N_{y}=\#\left\{y_{i}^{S}=y|i\right\}+\#\left\{y_{i}^{T}=y|i\right\} \end{array}$$
where *C* denotes the regularization parameter.

The term $\frac{1}{\#\left\{i|y_{i}^{S}=y\|y_{i}^{T}=y\right\}}\left(\sum_{y_{i}^{S}=y}P_{S}x_{i}^{S}\alpha_{i}^{S}+\sum_{y_{i}^{T}=y}P_{T}x_{i}^{T}\alpha_{i}^{T}\right)$ represents the mean of transferred samples with the label *y*, and the term $\frac{1}{N_{ST}}\left(\sum_{i}P_{S}x_{i}^{S}\alpha_{i}^{S}+\sum_{i}P_{T}x_{i}^{T}\alpha_{i}^{T}\right)$ represents the mean of all the samples. In this way, Equation (4) denotes the final optimization problem of JFSSS-HFT.

### 3.2. Solving the Optimization Problem of JFSSS-HFT

To solve the optimization problem of JFSSS-HFT, Equation (4) is rewritten in matrix form, as shown below.
(5)$$\begin{array}{l} \underset{P_{S},P_{T},\alpha_{i}^{S},\alpha_{i}^{T}}{\min}tr\left(PX\times diag\left(\alpha \right)\sum_{y=1}^{k}L_{y}diag{\left(\alpha \right)}^{T}\times X^{T}P^{T}\right)\\ s.t.PP^{T}=I\\ 0\leq \alpha \left(i\right)\leq 1\\ \sum_{i\leq N_{S}}\alpha \left(i\right)=\left(1-w_{S}\right)N_{S}\\ \sum_{i>N_{s}}\alpha \left(i\right)=\left(1-w_{T}\right)N_{T} \end{array}$$
where the operators diag(⋅) and $P=\left[P_{S},P_{T}\right]$ denote the diagonal matrix and the co-projection matrix, respectively. The variable $\alpha =\left[\alpha_{1}^{S},\ldots ,\alpha_{N_{S}}^{S},\alpha_{1}^{T},\ldots ,\alpha_{N_{S}}^{T}\right]$. $X=\left[\begin{array}{l} X_{S},0\\ 0,X_{T} \end{array}\right]$ represents the data matrix from the source domain and target domain. $L_{y}=\left[y_{temp}^{S,y},y_{temp}^{T,y}\right]\times {\left[y_{temp}^{S,y},y_{temp}^{T,y}\right]}^{T}-C\left[y_{temp2}^{S,y},y_{temp2}^{T,y}\right]\times {\left[y_{temp2}^{S,y},y_{temp2}^{T,y}\right]}^{T}$, where
(6)$$\begin{array}{l} y_{temp}^{S,y}\left(i\right)=\left\{ \begin{array}{l} \frac{1}{\#\left\{y_{j}^{S}=y|j\right\}}if\;y_{i}^{S}=y\\ 0\;otherwise \end{array}\right.\\ y_{temp}^{T,y}\left(i\right)=\left\{ \begin{array}{l} -\frac{1}{\#\left\{y_{j}^{T}=y|j\right\}}if\;y_{i}^{T}=y\\ 0\;otherwise \end{array}\right.\\ y_{temp2}^{S,y}\left(i\right)=\left\{ \begin{array}{l} \frac{1}{N_{y}}-\frac{1}{N_{ST}}if\;y_{i}^{S}=y\\ -\frac{1}{N_{ST}}\;otherwise \end{array}\right.\\ y_{temp2}^{T,y}\left(i\right)=\left\{ \begin{array}{l} \frac{1}{N_{y}}-\frac{1}{N_{ST}}if\;y_{i}^{T}=y\\ -\frac{1}{N_{ST}}\;otherwise \end{array}\right. \end{array}$$

It is noted that the optimization problem of JFSSS-HFT is nonconvex, and the variables are limited by multiple equality and inequality constraints. In order to solve the above optimization problem efficiently, the alternative-direction methods of multipliers (ADMM) [23] are introduced to alternately optimize the variables until the termination condition is satisfied. First, the auxiliary vector v∈ℝNS+NT is constructed and added to the optimization problem as follows.
(7)minPS,PT,αiS,αiTtr(PX×diag(α)∑y=1kLydiag(α)T×XTPT)s.t.PPT=Iα(i)=v(i)∑i≤NSα(i)=(1−wS)NS∑i>Nsα(i)=(1−wT)NT0≤v≤1

Using the Lagrangian multiplier, the augmented Lagrangian function [18] can be drawn as follows.
(8)ℒ(P,α,v,λ,λ1,λ2)=tr(PX×diag(α)∑y=1kLydiag(α)T×XTPT)+∑iλ(i)⋅(α(i)−ν(i))+λ1(∑i≤NSα(i)−(1−wS)NS)+λ2(∑i>Nsα(i)−(1−wT)NT)+μ2∑i‖α(i)−ν(i)‖F2+μ2‖∑i≤NSα(i)−(1−wS)NS‖F2+μ2‖∑i>Nsα(i)−(1−wT)NT‖F2s.t.PPT=I0≤v≤1
where λ∈ℝNS+NT, λ1, and λ2 denote the Lagrangian multipliers. The parameter μ>0 represents the penalty coefficient. The optimization processing is performed alternatively.

When optimizing the projection matrix P, the other variables are fixed. The current optimization problem can be considered a generalized eigenvalue problem, and the solution of P is the eigenvectors of $\sum_{y=1}^{k}X\times diag\left(\alpha \right)\sum_{y=1}^{k}L_{y}diag{\left(\alpha \right)}^{T}\times X^{T}$ corresponding to the *M* smallest eigenvalues.

When optimizing the adaptive weight factor α, the partial derivative of ℒ with respect to α is zero, and we have:(9)$$\begin{array}{l} \frac{\partial ℒ\left(P,\alpha ,v,\lambda ,\lambda_{1},\lambda_{2}\right)}{\partial \alpha }=0\Rightarrow \\ \alpha =\\ {\left(2\times \left(X^{T}P^{T}PX\right)\circ \left(\sum_{y=1}^{k}L_{y}\right)+\mu \times diag\left(e\right)+\mu \times \left(\left[e_{S},0_{N_{T}}\right]{\left[e_{S},0_{N_{T}}\right]}^{T}+\left[0_{N_{S}},e_{T}\right]{\left[0_{N_{S}},e_{T}\right]}^{T}\right)\right)}^{-1}\\ \left(\mu \times \left[\left(1-w_{S}\right)N_{S}\times e_{S},\left(1-w_{T}\right)N_{T}\times e_{T}\right]+\mu \times \nu -\lambda -\left[\lambda_{1}e_{S},\lambda_{2}e_{T}\right]\right). \end{array}$$

The operators ∘, $e_{S}$, and $0_{N_{S}}$ denote the Hadmard product (i.e., pair-wise product), the vector that all elements are equal to 1, and the zero vector with an $N_{S}$ dimension. According to Equation (9), α can be updated.

To update auxiliary variables v, the partial derivative of ℒ with respect to v is zero. It can be obtained with the following equation.
(10)$$\begin{array}{l} \frac{\partial ℒ\left(P,\alpha ,v,\lambda ,\lambda_{1},\lambda_{2}\right)}{\partial v}=0\Rightarrow \\ v=\alpha +\frac{1}{\mu }diag\left(\lambda \right) \end{array}$$

Combining this with the following inequality constraints, v can be updated as follows.
(11)$$v=\left\{ \begin{array}{l} v+\frac{1}{\mu }diag\left(\alpha \right)\;if\;0\leq v+\frac{1}{\mu }diag\left(\alpha \right)\leq 1\\ 0\;if\;v+\frac{1}{\mu }diag\left(\alpha \right)<0\\ 1\;if\;v+\frac{1}{\mu }diag\left(\alpha \right)>1 \end{array}\right.$$

To update the Lagrangian multipliers and penalty coefficient, the following update operation can be constructed.
(12)$$\begin{array}{l} \lambda \left(i\right)=\lambda \left(i\right)+\mu \left(\alpha \left(i\right)-\nu \left(i\right)\right)\\ \lambda_{1}=\lambda_{1}+\mu \left(\sum_{i\leq N_{S}}\alpha \left(i\right)-\left(1-w_{S}\right)N_{S}\right)\\ \lambda_{2}=\lambda_{2}+\mu \left(\sum_{i>N_{s}}\alpha \left(i\right)-\left(1-w_{T}\right)N_{T}\right)\\ \mu =\min\left(p\mu ,\mu_{\max}\right) \end{array}$$
where *p* and $\mu_{\max}$ denote the learning rate and the maximum penalty coefficient, respectively.

The optimization process repeats the above updating steps until the number of iterations exceeds the threshold or the difference between the variables in the *k*-th iteration and the variables in the *k*+1-th iteration is less than the threshold, i.e., the following convergence criterion is satisfied.
(13)‖[diag(α),v,λ,λ1,λ2]−[diag(α′),v′,λ′,λ′1,λ′2]‖F2≤δ1||kiter>δ2
where [diag(α′),v′,λ′,λ′1,λ′2] denotes the updated variables. The optimization solving algorithm of JFSSS-HFT is summarized in Algorithm 1.
**Algorithm 1.** JFSSS-HFT Optimization Algorithm.Input: samples from source domain XS, samples from target domain XT, and corresponding label.Output: P,αStep 1. Initialize P,α,v,λ,λ1,λ2,μ,p,μmaxwhile trueStep 2. update *P* using eigenvalue decomposition.Step 3. update α using Equation (9).Step 4. update v using Equation (11).Step 5. update λ1,λ2,μ using Equation (12).Step 6. check the convergence criteria. If the condition is met, break; otherwise, go to Step 2.end

## 4. Experiments and Analysis

To evaluate the quality of the proposed JFSSS-HFT effectively, two datasets containing RSIs of different resolutions are built and utilized to examine the performance of JFSSS-HFT compared with certain state-of-the-art feature transfer methods. The detailed information is described as follows.

Dataset (1) comprises four types of airplanes patches with different resolutions collected from the Google Earth service. Among them, 120 airplane patches with 0.5-m spatial resolution are regarded as samples from the target domain, and 200 airplane patches with 1-m spatial resolution are considered as samples from the source domain. Since these patches are acquired by different illumination conditions and contain airplanes with different orientations and different backgrounds, it can verify the performance of the proposed JFSSS-HFT for object classification using RSIs. The representative samples in dataset 1 are displayed in Figure 2.

Dataset (2) contains five types of ship patches with different resolutions collected from the HRSC-2016 publicly available dataset [24], where 120 ship patches with a 1.07 m spatial resolution are regarded as the target domain, and the 200 ship patches with a lower spatial resolution are considered as the source domain. Since these ship patches are observed under different illumination conditions, different ships present different sizes of occlusion due to their superstructure. The representative ship patches are displayed in Figure 3.

The experiments consist of three parts. In Section 4.1, the multiple features of patches with different resolutions are extracted and used for object classification. In Section 4.2, the convergence and main parameter settings of JFSSS-HFT are analyzed in detail. In Section 4.3, the performance of JFSSS-HFT is evaluated and compared with certain state-of-the-art feature transfer methods using dataset 1. Then, the quality of JFSSS-HFT is verified and compared with certain state-of-the-art feature transfer methods using dataset 2. All simulations are performed running on an i7-7700 Intel processor at 3.6 GHz and 8 GB memory with a Windows 10 system.

### 4.1. Extraction of Features for Multiresolution Object Patches

To ensure effective classification results, several typical image descriptors are utilized to extract the rich features of object patches. Detailed information of these typical image descriptors can be found in the following:1.Histogram of oriented gradient features

The histogram of oriented gradient (HoG) features [25] are widely used in different image-recognition tasks, which can describe the contour distribution within the patch in different positions. In the experiments, the size of the cell and the number of bins for HoG features are set to [4×4] and 9, respectively.

2.Local binary pattern features

The local binary pattern (LBP) [26] is the classical textural feature. In the experiments, the length of the radius is set to 8.

3.Gabor features

Gabor filters [27] can be used to obtain the time-frequency response of different positions in patches. In the experiments, four orientations and five scales of Gabor filters are constructed to extract the features of the patch.

The examples of features extraction results are displayed in Figure 4.

### 4.2. Analysis of the Convergence and the Main Parameter Setting of JFSSS-HFT

In this section, the convergence of the ADMM-based JFSSS-HFT optimization algorithm (i.e., Algorithm 1) is analyzed under different learning rates, i.e., *p*. The obtained convergence curves are shown in Figure 5. It can be seen that the objective function value (see Equation (5)) has a slight oscillation in the first few iterations, and the objective function values gradually decrease when the iteration number increases. More importantly, it is observed that the different convergence speeds can be obtained under different *p*. In addition, alongside the increase in *p*, a large convergence speed can be obtained. For different values of *p*, the objective function values become stable after 120 iterations. Therefore, it is reasonable to set the minimum iteration number to exceed 200.

The main parameters of the proposed JFSSS-HFT include *M*, *C*, $w_{S}$, and $w_{T}$, where *M*, *C*, $w_{S}$, and $w_{T}$ denote the dimension of sub-space, the regularization parameter, the pre-established proportion of outliers for the source domain, and the pre-established proportion of outliers for the target domain, respectively. To demonstrate the impact of different values of these parameters on the performance of JFSSS-HFT, the experiments are conducted on dataset 1 to obtain the classification results under different parameter settings. In detail, 120 samples from the target domain are divided into two groups of equal size. For the 200 samples from the source domain and the samples in the first group of the target domain, the transferred samples are obtained by applying JFSSS-HFT with different parameter settings, and the 1-nearest neighbor (1-NN) [28] classifier is used to examine the classification results using samples in the second group of the target domain. The obtained classification results are shown in Figure 6.

As can be seen from Figure 6, different classification results are obtained under different parameter settings. In detail, it is seen that the classification accuracy first improved and then decreased with the increase in *M*. The reason is that with the increase in dimensions for the transferred features, they will contain more information. Nevertheless, if the dimensions of features are too large, the transferred features may contain noise data. In addition, it was observed that the classification accuracy with *C* = 4 is obviously higher than the classification accuracy with the other values of *C*. Since the parameters $w_{S}$ and $w_{T}$ can be used to determine the rate of the outliers for samples, it is noted that the $w_{S}$ and $w_{T}$ with proper values (e.g., $w_{S}=0.1$ and $w_{T}=0.1$) can be used to improve the classification results. In contrast, overly large parameters will lead to the removal of high-quality samples and therefore weaken the classification results, e.g., $w_{S}=0.15$ and $w_{T}=0.15$. Moreover, it is noted that the best results are obtained under *M* = 250, *C* = 4, $w_{S}=0.1$, and $w_{T}=0.1$.

### 4.3. Evaluation of the Performance of the JFSSS-HFT Compared with Typical Transfer Methods

To evaluate the performance of the JFSSS-HFT compared with typical feature transfer methods, including principal component analysis (PCA) [29], transfer component analysis (TCA) [12], Subspace Alignment (SA) [30], joint probability adaptation (JDA) [13], and active transfer learning (ATL) [18], different proportions of samples in the target domain of dataset 1 together with samples in the source domain of dataset 1 are used as the training set and the remaining samples in the target domain of dataset 1 are used as the test set to obtain the classification results of different feature transfer methods. The image interpolation (II) method (i.e., bilinear interpolation) [31] can be considered a special feature transfer method to transform the image patches from the source domain to the same size of image patches from the target domain. Therefore, in addition to the above feature transfer methods, we also use the image interpolation method to process the samples and then train the classifier. All feature extractors described in Section 4.1 are used to extract the features of the input image, and the obtained features are then used to examine the performance of different feature transfer methods. To make a fair comparison, the dimension of transferred features for all methods is set to 250. In addition, all methods select the optimal parameter to obtain the final classification results. The obtained results are shown in Figure 7.

From Figure 7, it can be seen that each feature transfer method obtains higher accuracy with the increase in the size of the training set because a larger training set indicates better performance of the trained classifier. In addition, it can be found that II obtained the worst results compared with other methods. The reason is that the II method cannot eliminate the difference between the source domain and target domain effectively. Moreover, it is found the comparison methods, except II, obtain similar results. Among these comparison methods, ATL presents the best results when the proportion of training size is equal to 0.8, because ATL can reduce the impact of the outliers adaptively when learning the sub-space of parameters. This means ATL is robust to samples with poor quality. In comparison, since the proposed JFSSS-HFT is not only robust to the outliers but also adapts to heterogeneous data, the proposed JFSSS-HFT obtains the best results among all the methods, under different proportions of the training set. It indicates the quality of the proposed JFSSS-HFT for the image classification of RSIs with different resolutions.

Subsequently, the performance of JFSSS-HFT is further verified using dataset 2. Similar to the process for dataset 1, we compare the test accuracy of different feature transfer methods using different proportions of the training set, and the obtained results are shown in Figure 8. Figure 8 shows the average accuracy of using different proportions of the training set for different feature transfer methods. It is seen that different methods obtain different accuracies under different sizes of the training set. Specifically, in comparison to Figure 7, it can be seen that the classification accuracy of dataset 2 is smaller than that of dataset 1, because the ship patches contain more interference (e.g., the superstructure in the ship) than airplane patches. The II method obtains the worst results because the interpolation method find it difficult to eliminate the difference between different domains. The PCA method obtains the second-worst results because it did not take into account the domain distribution differences. In addition, since ATL and the proposed JFSSS-HFT method can reduce the impact of the outliers adaptively, it was observed that the ATL method and the proposed JFSSS-HFT method obtain relatively effective results. Note that SA only considers the spatial alignment in the adaptive domain problem. JDA minimizes the marginal and conditional distribution differences between domains, while TCA can only minimize the marginal differences. In comparison, our method not only performs probabilistic adaptation but also penalizes the outlier factor, so its effect is the greatest. Therefore, it can be seen that JFSSS-HFT achieves the best results among all the methods. This indicates the quality of the proposed JFSSS-HFT method for image classification using RSIs with different resolutions.

Furthermore, considering that deep learning methods have been widely used in image classification recently, experiments were conducted on dataset 1 and dataset 2 to further compare the proposed method with deep learning methods, i.e., the ResNet-18 [32] method and the scale-free convolutional neural network (SF-CNN) [33]. For ResNet-18, the image interpolation method was utilized in preprocessing to adjust the input with the same size. Regarding SF-CNN, it can process the input with different sizes (i.e., RSIs with different resolutions) naturally. To facilitate a fair comparison, both the deep learning methods and the proposed JFSSS-HFT method adopt the same training set and test set. In detail, for dataset 1 or dataset 2, the training set contains half of the samples from the target domain and all the samples from the source domain, and the test set contains the remaining samples from the target domain. In addition, the batch size and learning rate are set to 2 and $5\times 10^{-4}$, respectively. The obtained results are shown in Table 1.

From Table 1, it is found that the SF-CNN method obtains more accurate classification results than ResNet-18, because SF-CNN can effectively extract information embedded in the image with different resolutions. Nevertheless, JFSSS-HFT outperforms these two deep learning methods for airplane classification and ship classification. The reason is that the deep learning methods rely heavily on a large number of training samples, while the proposed JFSSS-HFT method is able to achieve better results with fewer training samples. This finding further demonstrates the quality of the proposed JFSSS-HFT method for object classification using RSIs with different resolutions.

## 5. Conclusions

To improve the classification results for RSIs with high-resolution, the JFSSS-HFT method is proposed to map the features of RSIs with different resolutions to the common feature space and reduce the impact of outliers adaptively, and then jointly train the classifier to obtain the effective classification results. Compared with the existing feature transfer methods that only focus on feature mapping for the feature space, the proposed JFSSS-HFT method can adapt to the heterogeneous data by considering the feature space and sample space simultaneously, reducing the occurrence of negative transfer. Experimental results demonstrate the proposed JFSSS-HFT method outperforms the typical feature transfer methods under two datasets containing airplane patches and ship patches with different resolutions.

Future work will focus on extending the JFSSS-HFT to a semi-supervised version and an unsupervised version to improve the applicability of the method.

## Figures and Tables

**Figure 1 sensors-21-07568-f001:**
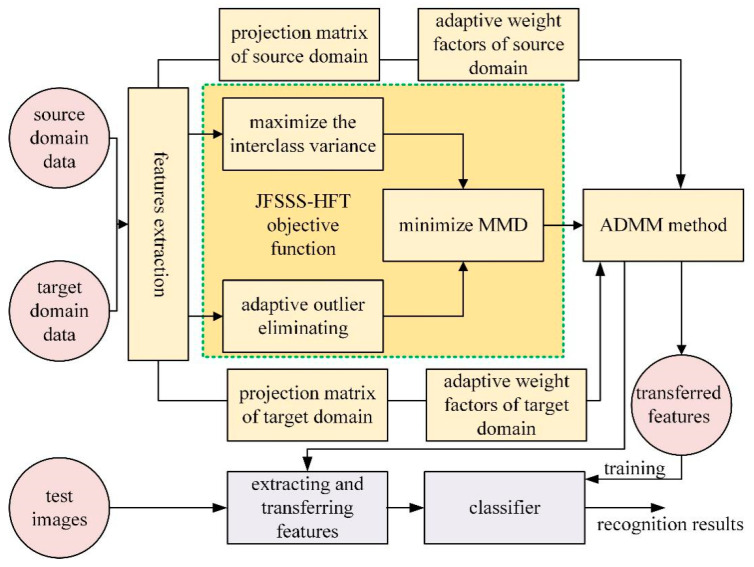
Illustration of the procedure of the proposed JFSSS-HFT for object recognition using RSIs from target domain and source domain.

**Figure 2 sensors-21-07568-f002:**
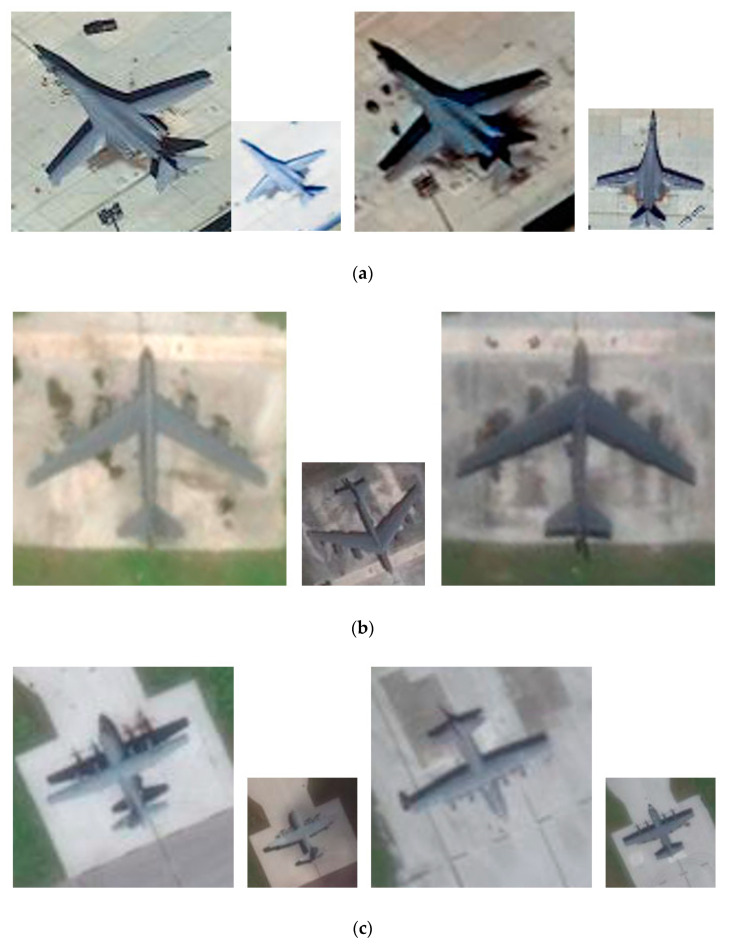
Representative samples from dataset 1. (**a**) Airplanes with type 1 from target domain and source domain. (**b**) Airplanes with type 2 from target domain and source domain. (**c**) Airplanes with type 3 from target domain and source domain. (**d**) Airplanes with type 4 from target domain and source domain.

**Figure 3 sensors-21-07568-f003:**
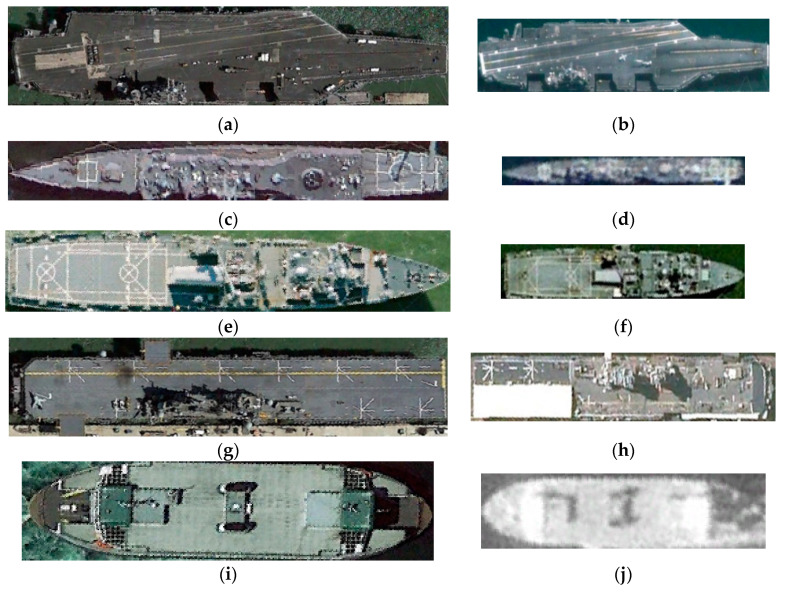
Representative samples from dataset 2. (**a**) Ship with type 1 from target domain. (**b**) Ship with type 1 from source domain. (**c**) Ship with type 2 from target domain. (**d**) Ship with type 2 from source domain. (**e**) Ship with type 3 from target domain. (**f**) Ship with type 3 from source domain. (**g**) Ship with type 4 from target domain. (**h**) Ship with type 4 from source domain. (**i**) Ship with type 5 from target domain. (**j**) Ship with type 5 from source domain.

**Figure 4 sensors-21-07568-f004:**
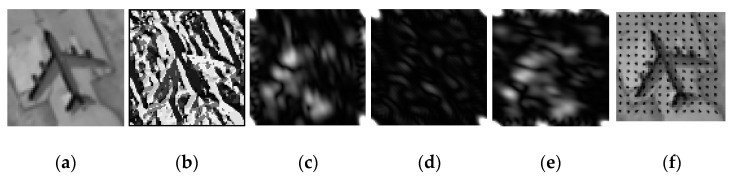
Multiple feature visualization results for airplane patch, where (**a**) denotes the original patch. The local binary pattern features (LBP) are displayed in (**b**). (**c**–**e**) denote the Gabor feature using Gabor filter with different orientations and scales, and (**f**) denotes the HoG feature.

**Figure 5 sensors-21-07568-f005:**
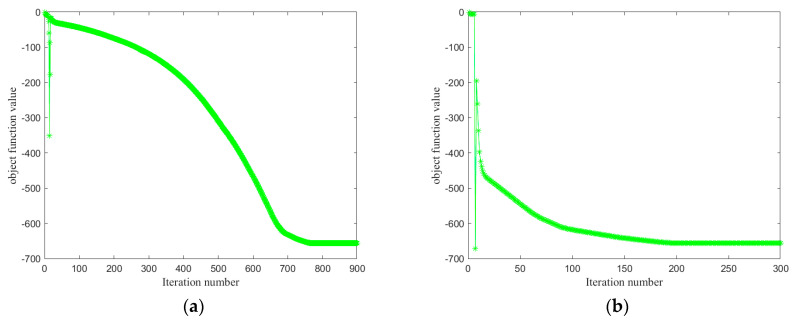
The convergence of JFSSS-HFT under different *p.* (**a**) The convergence of JFSSS-HFT under *p* = 1.2. (**b**) The convergence of JFSSS-HFT under *p* = 1.5. (**c**) The convergence of JFSSS-HFT under *p* = 2.

**Figure 6 sensors-21-07568-f006:**
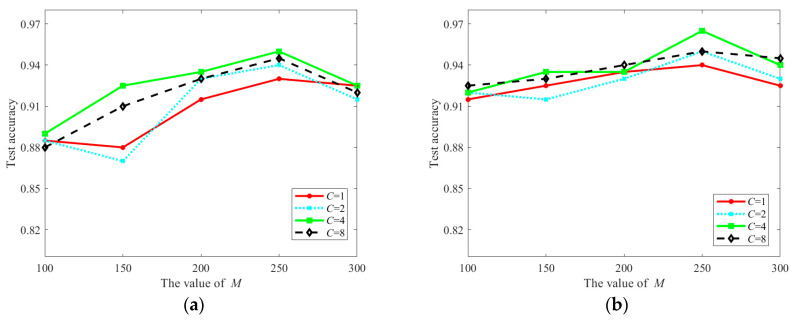
Classification results of JFSSS-HFT under different parameter settings. (**a**) Classification results under $w_{S}=0.05$ and $w_{T}=0.05$. (**b**) Classification results o under $w_{S}=0.1$ and $w_{T}=0.1$. (**c**) Classification results of JFSSS-HFT for different parameter settings under $w_{S}=0.15$ and $w_{T}=0.15$.

**Figure 7 sensors-21-07568-f007:**
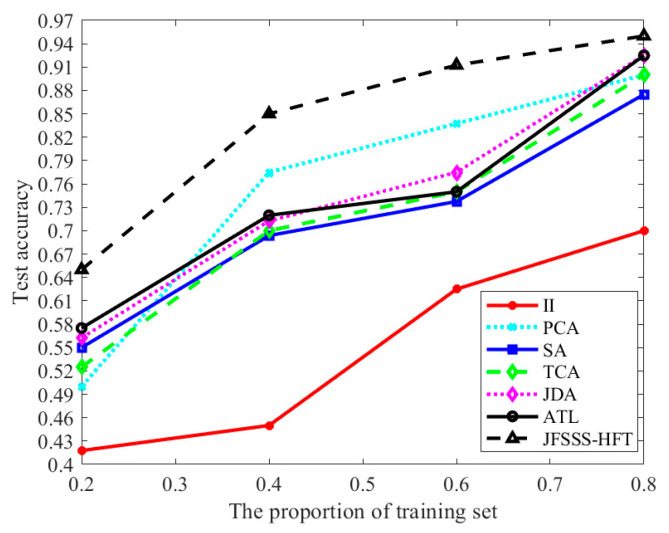
Classification results of different features transfer methods using training set with different sizes of dataset 1.

**Figure 8 sensors-21-07568-f008:**
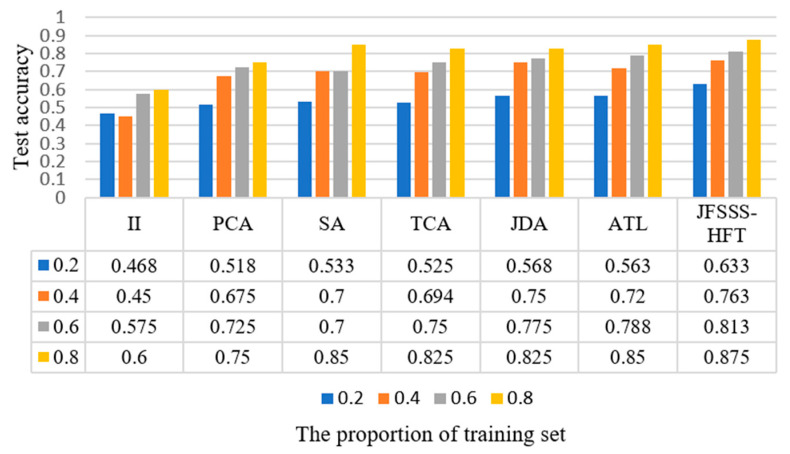
Classification results of different features transfer methods using training set with different sizes of dataset 2.

**Table 1 sensors-21-07568-t001:** Classification results of the proposed features transfer methods compared with deep learning methods.

	ResNet-18	SF-CNN	JFSSS-HFT
airplane classification	68.3%	70%	81.7%
ship classification	66.7%	71.6%	80%

## Data Availability

Not applicable.

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
