# Peer review of "Joint Feature-Space and Sample-Space Based Heterogeneous Feature Transfer Method for Object Recognition Using Remote Sensing Images with Different Spatial Resolutions"

_sensors, 2021, doi:10.3390/s21227568_

Round 1

Reviewer 1 Report

The paper proposed a new transfer learning algorithm for the classification of remotely-sensed data with different spatial resolutions. In general, the paper is well-written and well-structured. Nevertheless, the authors should address some issues before the possible publication of the paper.

  • Sample space is usually inferred as the statistical sample space, but this definition does not fit the use of this term within the paper. So, what do the authors mean by the sample space? ; and how do they define this space?
  • Throughout the paper different concepts, such as different dimensions, different resolutions, have been used to describe the heterogeneous feature space. Although these differences can be considered the reason for two feature spaces to be heterogeneous, the authors should clearly define the heterogeneous feature spaces in their paper.
  • The proposed algorithm claimed to remove outliers. However, it is not clear how.
  • Please provide a complete introduction to the used data sets. This introduction should include information about the number of labeled samples (training, validation) for each domain, the dimensionality (particularly the number of spectral bands) of each data set, the spatial resolutions.
  • It is not mentioned which classifier is used to classify the final set of extracted features.
  • Most of the recent researches on transfer learning focused on the deep learning method. It could be very informative for the readers to know how accurate is the proposed method compared with the deep learning methods.
  • As the final comment, I suggest using the more common phrase “Image patch” instead of the “image slice”

Author Response

Thank you for your time and patience. It is my honor to answer your questions about our paper.

Response to the comments from reviewer 1

This paper proposed a new transfer learning algorithm for the classification of remotely-sensed data with different spatial resolutions. In general, the paper is well-written and well-structured. Nevertheless, the authors should address some issues before the possible publication of the paper.

Comment 1:

  • Sample space is usually inferred as the statistical sample space, but this definition does not fit the use of this term within the paper. So, what do the authors mean by the sample space? ; and how do they define this space?

Reply: Thank you very much for your comment and valuable suggestions. In the probability theory, the sample space is defined as a set of all possible outcomes or results of the experiments. A sample space is described as a set , where element  and N denote the i-th possible outcomes or experiment results and the number of possible outcomes or experiment results, respectively. Different from the definition of sample space in probability theory, the sample space in our paper is defined as a set , where element  and N denote the i-th sample and the number of samples, respectively. For the proposed JFSSS-HFT, it selects adaptively useful samples and reduces the impact of outliers from sample space(i.e., the sample set). To emphasize the meaning of “sample space” used throughout this paper, we have added the definition of sample space in the introduction of the revised manuscript.

Comment 2:

  • Throughout the paper different concepts, such as different dimensions, different resolutions, have been used to describe the heterogeneous feature space. Although these differences can be considered the reason for two feature spaces to be heterogeneous, the authors should clearly define the heterogeneous feature spaces in their paper.

Reply: Thank you for your constructive suggestion. To explain the meaning of heterogeneous feature space clearly, we have added the definition of heterogeneous feature space in the revised manuscript, as shown below.

Definition 1 (heterogeneous features and heterogeneous feature transfer) Given two features describing objects, if these two features present the different dimensions, they are considered as heterogeneous features. In addition, if features from source domain and features from target domain present different dimensions, the corresponding transfer learning method is regarded as heterogeneous feature transfer.”

Comment 3:

  • The proposed algorithm claimed to remove outliers. However, it is not clear how.

Reply: Thank you for your comment. The outliers in the proposed JFSSS-HFT can be considered as the samples that may interfere with the results of feature transfer. In the previous manuscript, we claimed that our JFSSS-HFT can remove outliers, while this statement is not rigorous. Therefore, the “remove outliers” is modified as “reduce the impact of outliers”. Actually, the proposed JFSSS-HFT can reduce the impact of outliers adaptively during the optimization solving, as shown in Eq.(5). The bound constraints indicate the range of  and  is from 0 to 1.  If  weight factor is close to 0, the corresponding samples are considered as outliers and its effect on the transfer results is weaken. During the optimization of Eq.(5) using ADMM, the proper values of  and  are assigned adaptively to each sample from the source domain and target domain to reduce the objective function value as much as possible. In this way, if the sample is unfavorable to transfer, it is given a small weight to reduce its effect on the transfer results.

Comment 4:

  • Please provide a complete introduction to the used data sets. This introduction should include information about the number of labeled samples (training, validation) for each domain, the dimensionality (particularly the number of spectral bands) of each data set, the spatial resolutions.

Reply: Thank you for your kindly and detailed comment. In the revised manuscript, the detailed information of the used datasets is given in Section.4, as shown in the following:

“Dataset 1) dataset 1 comprises 4 types of labeled airplanes slices with different resolutions collected from Google Earth service. All these slices contain three spectral bands, i.e., R, G, and B bands. Among them, 120 airplanes slices with 0.5-meter spatial resolution are regarded as samples from target domain, and 200 airplanes slices with 1-meter spatial resolution are considered as samples from source domain. Since these slices are acquired by different illumination conditions and contain airplanes with different orientations and different backgrounds, it can verify the performance of the proposed JFSSS-HFT for object classification using RSIs. The representative samples in dataset 1 are displayed in Figure 2.

Dataset 2) dataset 2 contains 5 types of labeled ship slices with different spatial resolutions collected from HRSC-2016 publicly available dataset [22], where 120 ship slices with 1.07 meters spatial resolution are regarded as the target domain, and the 200 ship slices with the lower spatial resolution (1.07 meters to 2 meters) are considered as the source domain. All the ship slices contain three spectral bands, i.e., R, G, and B bands. Since these ship slices are observed under different illumination conditions, different ships present different sizes of occlusion due to their superstructure. The representative ship slices are displayed in Figure 3.”

         In addition, to valid effectively the performance of the JFSSS-HFT, training and test sets of different sizes are divided from the dataset for comparing the proposed method with typical feature transfer methods. Therefore, we did not define a specific number of training samples versus test samples in the dataset description.

Comment 5:

  • It is not mentioned which classifier is used to classify the final set of extracted features.

Reply: Thank you for your detailed comment. The used classifier is the 1-nearest neighbor (1-NN) classifier, and we have emphasized the used classifier in the revised manuscript.

Comment 6:

  • Most of the recent researches on transfer learning focused on the deep learning method. It could be very informative for the readers to know how accurate is the proposed method compared with the deep learning methods.

Reply: Thank you for your constructive suggestion. In addition, to further evaluate the quality of the JFSSS-HFT, the supplementary experiments are added in the revised manuscript to compare the JFSSS-HFT with the deep learning methods, as described below.

“Furthermore, considering that deep learning methods have been widely used in image classification recently. The experiments are conducted on dataset 1 and dataset 2 to further compare the proposed method with the deep learning methods, i.e., the ResNet-18 [32] method and scale-free convolutional neural network (SF-CNN) [33]. For ResNet-18, the image interpolation method is utilized as preprocessing to adjust input with the same size. For SF-CNN, it can process the input with different sizes (i.e., RSIs with different resolutions) naturally. To facilitate a fair comparison, both the deep learning methods and the proposed JFSSS-HFT adopt the same training set and test set. In detail, for dataset 1 or dataset 2, the training set contains half of samples from target domain and all the samples from source domain, and the test set contains the remaining samples from target domain. In addition, the batch size and the learning rate are set to 2 and , respectively. The obtained results are shown in Table 1.

Table 1. Classification results of the proposed features transfer methods compared with deep learning methods

ResNet-18

SF-CNN

JFSSS-HFT

airplane classification

68.3%

70%

81.7%

ship classification

66.7%

71.6%

80%

From Table 1, it is found that the SF-CNN obtains more accurate classification results than ResNet-18, because the SF-CNN can extract information embedded in the image with different resolutions effectively. Nevertheless, the JFSSS-HFT outperforms these two deep learning methods for airplane classification and ship classification. The reason is that the deep learning methods rely heavily on a large number of training samples, while the proposed JFSSS-HFT is able to achieve better results with fewer training samples. This finding further demonstrates the quality of the proposed JFSSS-HFT for object classification using RSIs with different resolutions.”

Comment 7:

  • As the final comment, I suggest using the more common phrase “Image patch” instead of the “image slice”

Reply: Thank you for your detailed comment. In the revised manuscript, we have replaced “image slice” with “Image patch”.

Reviewer 2 Report

Dear authors, see the attached pdf file for the comments.

Reviewer 3 Report

This paper proposed JFSSS-HFT for the high-resolution remote sensing images classification task. It constructed two matrixes to transfer features of samples with different resolutions to a common space. Besides, it used two learnable parameters to weight the contribution of training samples from both the source and the target domains. The comments are as follows:

1) The innovation needs to be clarified  clearer. The main innovation of this paper is weighting the contributions of training samples from both the source and target domains with two learnable parameters. However, the weighting method is the same as [17] which just weights the training samples of the source domain.

2)What’s the difference of constructing two projecting matrices in this paper with that of two projecting matrices for the source data and the target data proposed in [14].

3) The motivation is that the proposed JFSSS-HFT considers the difference between samples-space between the source domain and target domain. However, why is this consideration so important?

4) The description of the method is not accurate enough. For example, the authors said that JFSSS-HFT removed the outliers from the sample space adaptively to avoid the occurrence of the negative transfer. However, in fact, it is implemented by giving different weights.

5) Experimental verification is insufficient. The experimental dataset is too small to fully verify the algorithm. The deep-learning-based algorithms are suggested to be compared.

Round 2

Reviewer 1 Report

I appreciate the authors for applying most of my comments. However, there are yet some minor issues, which I listed here:

  • Page 2, line 54, the font of Y is different in this line.
  • Please double-check the paper, in some cases, the image slice is used instead of the image patch.
  • On page 5, the line spacing seems to be different in the last paragraph.
  • Page 6, the equation (1) and some explanation of the MMD algorithm are presented before, for sake of brevity please remove one of them.
  • Page 16, please write the title on page 7, where the text starts.
  • Page 17, The table in this image has low resolution. Please provide a higher-resolution image.

Reviewer 3 Report

The paper has been revised according to the comments.

Author Response

Thank you very much for your comment. We have checked throughout the revised manuscript to correct grammatical or spelling errors.